# DS2TA: Denoising Spiking Transformer with Attenuated Spatiotemporal Attention

## Abstract

Vision Transformers (ViT) are current high-performance models of choice for various vision applications. Recent developments have given rise to biologically inspired spiking transformers that thrive in ultra-low power operations on neuromorphic hardware, however, without fully unlocking the potential of spiking neural networks. We introduce `DS2TA`, a Denoising Spiking transformer with attenuated SpatioTemporal Attention, designed specifically for vision applications. `DS2TA` introduces a new spiking attenuated spatiotemporal attention mechanism that considers input firing correlations occurring in both time and space, thereby fully harnessing the computational power of spiking neurons at the core of the transformer architecture. Importantly, `DS2TA` facilitates parameter-efficient spatiotemporal attention computation without introducing extra weights. `DS2TA` employs efficient hashmap-based nonlinear spiking attention denoisers to enhance the robustness and expressive power of spiking attention maps. `DS2TA` demonstrates state-of-the-art performances on several widely adopted static image and dynamic neuromorphic datasets. Operated over 4 time steps, `DS2TA` achieves 94.92% top-1 accuracy on CIFAR10 and 77.47% top-1 accuracy on CIFAR100, as well as 79.1% and 94.44% on CIFAR10-DVS and DVS-Gesture using 10 time steps.

## 1 Introduction

Originally designed for natural language processing applications Vaswani et al. (2017), transformers have gained popularity in various computer vision tasks, including image classification Dosovitskiy et al. (2021), object detection (Carion et al., 2020), and semantic segmentation Xie et al. (2021). As a key mechanism of transformers, self-attention selectively focuses on relevant information, enabling capturing of long-range interdependent features.

Spiking neural networks (SNNs) are more biologically plausible than their non-spiking artificial neural network (ANN) counterparts (Gerstner & Kistler, 2002). Notably, SNNs can harness powerful temporal coding, facilitate spatiotemporal computation based on binary activations, and achieve ultra-low energy dissipation on dedicated neuromorphic hardware (Furber et al., 2014; Davies et al., 2018; Lee et al., 2022). The recent emergence of spiking transformer architectures represents a logical progression (Zhou et al., 2023; Zhang et al., 2022; Zhu et al., 2023). Particularly, promising results have been demonstrated by the spiking transformers of Zhou et al. (2023), which incorporate a spike-based self-attention mechanism. This self attention captures correlations between spatial input patches occurring at the same time point, which we refer to as "spatial-only" attention.

Motivated by the recent progress on spiking transformers, this work proposes a new architecture called Denoising Spiking transformer with Attenuated SpatioTemporal Attention (`DS2TA`). `DS2TA` enables fully-fledged spiking temporally attenuated spatiotemporal attention (TASA) as opposed to "spatial-only" attention of Zhou et al. (2023). TASA computes spiking queries, keys, values, and the final output of each attention block while taking into account correlations in input firing activities occurring in both time and space. Thus, it fully exploits the spatiotemporal computing power of spiking neurons for forming attentions, which are at the core of any transformer. Equally importantly, we facilitate parameter-efficient spatiotemporal attention

computation and employ nonlinear denoising to enhance the robustness and expressive power of spiking attention mechanisms, thereby boosting the overall model's performance.

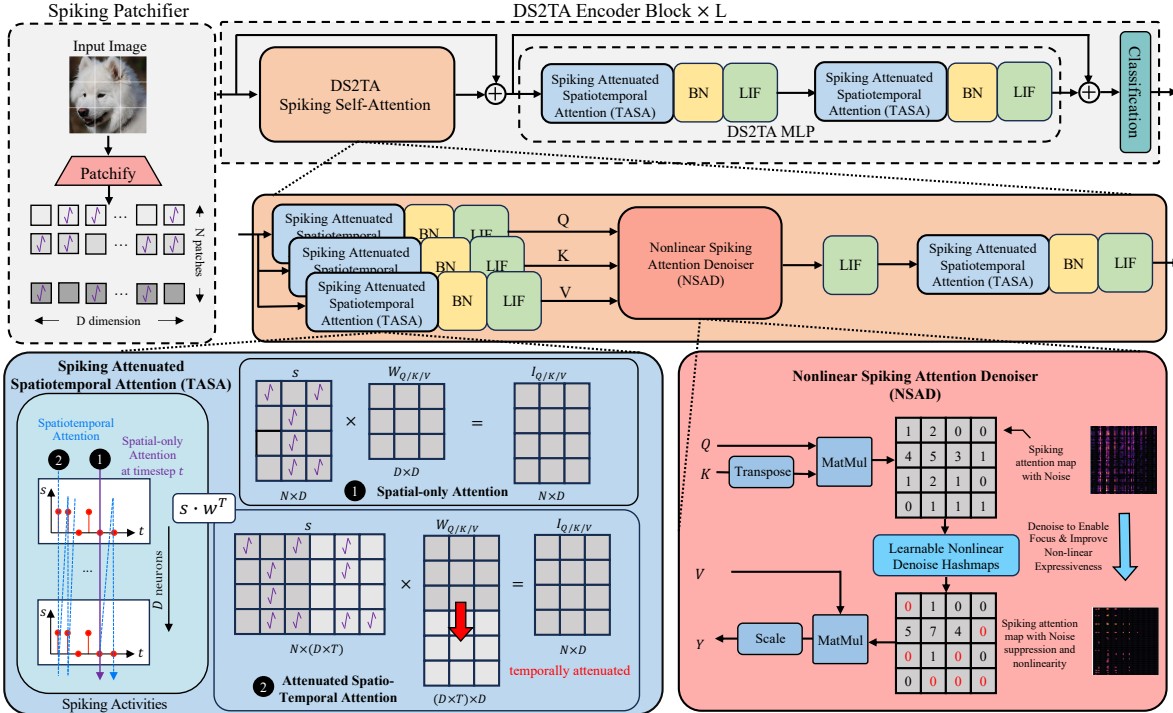

Figure 1: The overview of denoising spiking transformer with intrinsic plasticity and spatiotemporal attention: `DS2TA`.

The proposed `DS2TA` spiking transformer introduces several key contributions:

**Departure from "Spatial-Only" Attention**: In contrast to existing spiking transformers that exhibit "spatial-only" attention, such as those by Zhou et al. (2023), `DS2TA` moves beyond this limitation. It introduces temporally attenuated spatiotemporal attention (TASA) that considers not only spatial but also temporal correlations in input firings when computing queries, keys, values and final output of each attention block, providing a more comprehensive approach to attention.

**Parameter-Efficient Spatiotemporal Attention Computation**: `DS2TA` is designed to facilitate parameter-efficient computation of spatiotemporal attention via a technique called Attenuated Temporal Weight Replica. This approach dramatically reduces the number of temporally-dependent synaptic weights employed in TASA. This efficiency contributes to the overall optimization of the transformer, enhancing its scalability and resource utilization.

**Nonlinear Spiking Attention Denoisers (NSAD)**: `DS2TA` utilizes efficient hashmap-based nonlinear denoisers with learnable nonlinearity to enhance the robustness and expressive power of spatiotemporal attention maps, thereby further improving performance.

Extensive experimental evaluations on various static image and dynamic neuromorphic datasets, consistently demonstrate the superior performance of the `DS2TA` architecture in comparison to prior spiking transformer approaches.

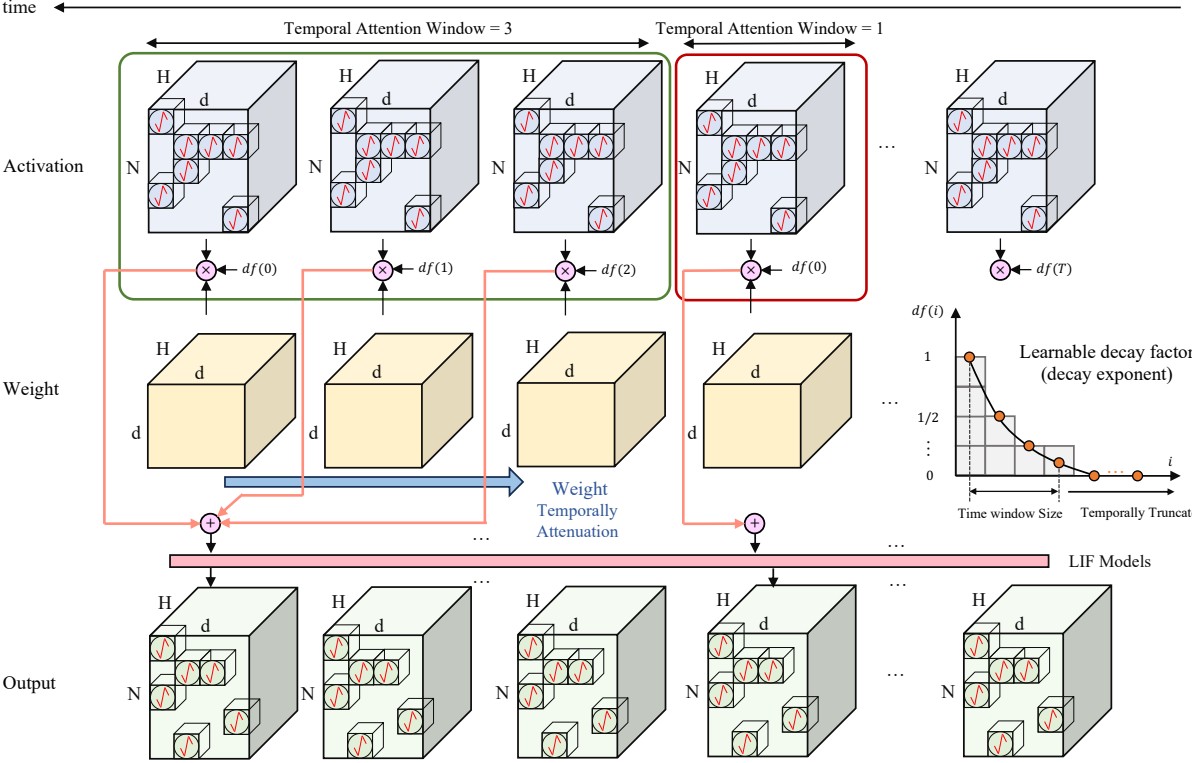

Figure 2: Proposed Temporally Attenuated Spatiotemporal Attention.

## 2    Related Work

### 2.1    Spiking Neural Networks

In the brain, computations typically occur in a spatiotemporal fashion Buzsaki (2006). As a computational model inspired by the brain, spiking neural networks (SNNs) are well suited for processing spatiotemporal information Maass (1997). Contrary to traditional deep learning models in ANNs, which relay information between layers through continuous values, SNNs operate on binary spikes across multiple time steps. To address non-differentiability of spiking activities, activation-based (or surrogate gradient) (Zenke & Ganguli, 2018; Li et al., 2021) and timing-based backpropagation training methods Shrestha & Orchard (2018); Zhang & Li (2020); Yang et al. (2021) and their combination have been proposed (Kim et al., 2020). Fang et al. (2021b) enhanced spiking neural networks by incorporating learnable membrane time constants. Zheng et al. (2020) introduced a threshold-dependent batch normalization method (STBP-tdBN) for directly training deep SNNs. Duan et al. (2022) proposed Temporal Effective Batch Normalization (TEBN) for enhancing training efficiency by rescaling the presynaptic inputs with different weights at every timestep. Yao et al. (2021) introduced a Temporal-wise Attention SNNs (TA-SNN) model that proposes a method of assigning significance to frames during training to reduce timesteps on DVS datastreams.

### 2.2    Spiking Transformers

Transformers Vaswani et al. (2023) and their variants Han et al. (2023); Khan et al. (2022); Lin et al. (2022) have emerged as a powerful model in deep learning based on the non-spiking artificial neural network (ANN) implementation. However, spiking-based transformers have not been well explored. Several recent studies have undertaken investigations into transformer-based spiking neural networks for tasks such as image classification

(Zhou et al., 2023), object tracking (Zhang et al., 2022), and the utilization of large language models (LLMs) (Zhu et al., 2023). Specifically, Zhang et al. (2022) introduced a non-spiking ANN based transformer designed to process spiking data generated by Dynamic Vision Sensors (DVS) cameras. On the other hand, Zhou et al. (2023) and Yao et al. (2023) have proposed a spiking vision transformer while incorporating only spatial and linear self-attention mechanisms. Wang et al. (2023) considered pairwise similarity between queries and keys in time and space. Zhu et al. (2023) developed an SNN-ANN fusion language model, integrating a transformer-based spiking encoder with an ANN-based GPT-2 decoder to enhance the operational efficiency of LLMs.

However, while the aforementioned spiking transformer models have made significant contributions to the field, they have not fully explored the potential of spatiotemporal self-attention mechanisms or addressed the issues of noise suppression and nonlinear scaling of attention maps, which are central to the research presented in this work.

## 3 Method

The conventional non-spiking Vision Transformer (ViT) architecture (Dosovitskiy et al., 2021) consists of patch-splitting modules, encoder blocks, and linear classification heads. Each encoder block includes a self-attention layer and a multi-layer perceptron (MLP) layer. Self-attention empowers ViT to capture global dependencies among image patches, thereby enhancing feature representation (Katharopoulos et al., 2020).

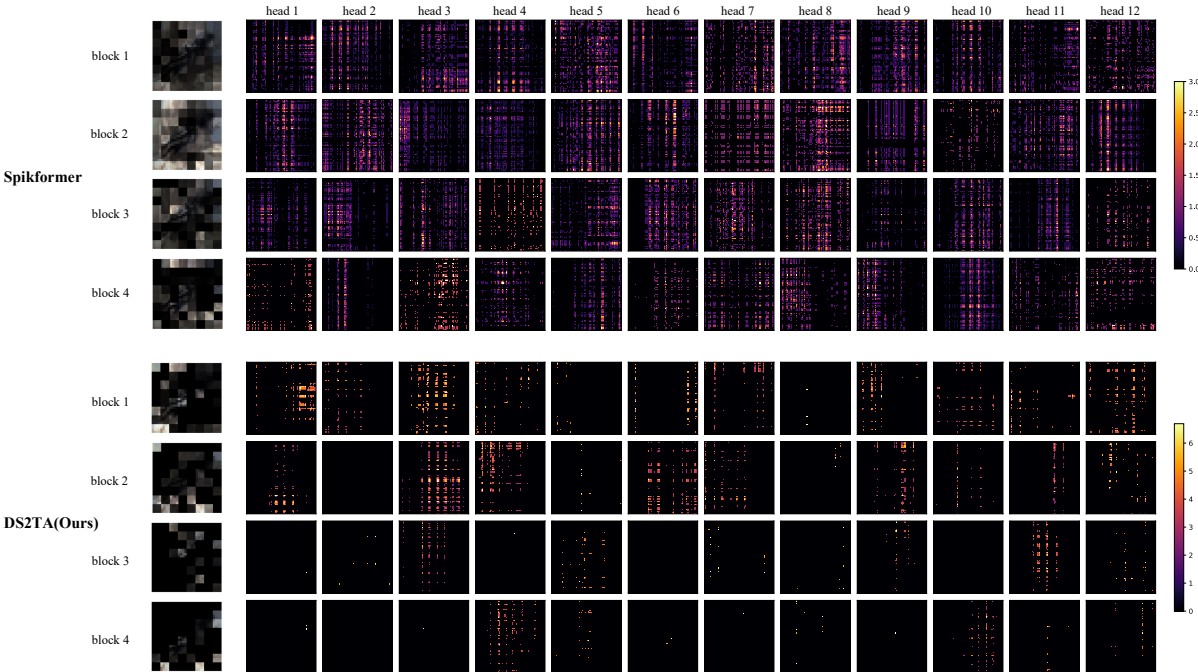

Figure 3: A visualization of attention maps across multiple heads horizontally and across multiple blocks vertically at the first timestep; Figures on the leftmost column[1]show the focus regions mapped onto an original image from CIFAR-100.

Recent spiking transformer models have adapted the architecture of ViT by incorporating a spiking patch-splitting module and processing feature maps of dimensions $N \times D$ over $T$ time steps using spiking neurons (Zhou et al., 2023). At their core, these models utilize spike-based multiplication to compute spatial-only attention maps in the form of $Q[t]K^T[t]$ for each time step, replacing the non-spiking counterparts $QK^T$ in the original ViT, where $Q$ and $K$ represent "query" and "key" respectively.

---

[1]The original image is masked based on the attention map of the last head at the first timestep.

The proposed `DS2TA` spiking transformer architecture introduces several innovations, as illustrated in Figure 1. The core `DS2TA` architecture consists of $L$ `DS2TA` multi-layer encoder blocks. Instead of computing spatial-only attentions as in (Zhou et al., 2023), we introduce a spatiotemporal spiking attention mechanism based on forming queries (Q), keys (K), values (V), and final output of each attention block while taking into account correlations in input firing activities across both time and space. Furthermore, we propose multi-head hashmap-based Nonlinear Spiking Attention Denoisers (NSAD) to suppress noise and increase expressive power for a given attention map via a learnable nonlinear mapping $f$, resulting in a denoised map $A = f(Q[t]K^T[t])$. The product of $A$ and the spike-based value $V$ is then passed to the subsequent layer in the encoder block.

## 3.1 Spiking Temporally Attenuated Spatiotemporal Attention (TASA)

As depicted in Figure 1, we apply the same spatiotemporal mechanism to two different layers within each encoder block. We elaborate on how this scheme is utilized to calculate the spike inputs for the three LIF spiking neuron arrays, whose output activations define the query $(Q)$, key $(K)$, and value $(V)$. The same spatiotemporal mechanism is adopted for computing the final output of each attention block.

To compute a "spatial-only" attention map (Zhou et al., 2023), as illustrated in ❶ of Figure 1, at each time point $t$, the spike inputs to the $l$-th encoder block, which are the outputs $s^{(l-1)}[t]$ of the previous $(l-1)$-th encoder block, undergo multiplication with corresponding weights to yield the inputs $I_{Q/K/V}[t]$ to the query/key/value LIF neuron arrays: $I_{Q/K/V}[t] = s^{(l-1)}[t] \times W_{Q/K/V}$, where $I_{Q/K/V}[t]$ and $s^{(l-1)}[t] \in R^{N \times D}$, and $W_{Q/K/V} \in R^{D \times D}$. $I_{Q/K/V}[t]$ only retains the information of spike outputs of $D$ neurons from the $(l-1)$-th encoder block at time $t$.

Notably, in `DS2TA`, we extend the attention from "spatial-only" to "spatiotemporal," as illustrated in ❷ of Figure 1, where not only the spiking activities of these $D$ neurons at time $t$ but also those occurring before $t$ are attended. This new mechanism allows `DS2TA` to attend to dependencies taking place in both time and space, and provides a means for fully exploring the spatitemporal computing power of spiking neurons under the context of transformer models, as shown in Figure 2.

### 3.1.1 Temporal Attention Window

The spiking spatio-temproal attention is confined within a Temporal Attention Window (TAW) to limit computational complexity.

Specifically, the input to the query/key/value neuron at location $(i, k)$ in block $l$ is based upon the firing activations of $D$ output neurons from the prior $(l-1)$-th block that fall under a given TAW $T_{AW}$:

$$I_{ik}^{(l)}[t] = \sum_{m=t-T_{AW}+1}^{t} \sum_{j=0}^{D} w_{(kj)(m)}^{(l)} s_j^{(l-1)}[m], \tag{1}$$

where $w_{(kj)(m)}^{(l)}$ is the temporally-attenuated synaptic weight specifying the efficacy of a spike evoked by the $j$-th output neuron of block $(l-1)$ $m$ time-steps before on the neuron at location $(i, k)$ in block $l$. With the above synaptic input, each query/key/value neuron is emulated by the following discretized leaky integrate-and-fire dynamics (Gerstner & Kistler, 2002):

$$V_{ik}^{(l)}[t] = (1 - \frac{1}{\tau_m})V_{ik}^{(l)}[t-1](1 - s_{ik}^{(l)}[t-1]) + I_{ik}^{(l)}[t], \tag{2}$$

$$s_{ik}^{(l)}[t] = H(V_{ik}^l[t] - \theta). \tag{3}$$

### 3.1.2 Attenuated Temporal weight replica

The spatiotemporal attention in Eq. 1 involves $T_{AW}$ temporally-dependent weights $w_{(kj)(m)}^{(l)}$, $m \in [0..T_{AW}]$ for a pair of presynaptic and postsynaptic neurons. We introduce a learnable scheme, called attenuated temporal weight replica, to reduce the number of temporally-dependent weights by a factor of $T_{AW}$. This amounts to set $w_{(kj)(m)}^{(l)}$, $m \in [1..T_{AW}]$, to be a temporally decayed value of $w_{(kj)(0)}^{(l)}$:

$$w_{(kj)(m)}^{(l)} = \mathbb{1}(m \geq 0) w_{(kj)(0)}^{(l)} df(m). \tag{4}$$

Here, as shown in Fig 2, $df(m)$ is the decay factor for $w_{(ik)(jm)}^{(l)}$. We make all decay factors a power-of-two, which can be efficiently implemented by low-cost shift operations:

$$df(m) = 2^{-\tau^{(l)} m}, \tag{5}$$

where $\tau^{(l)}$ is a layer-wise learnable integer decay exponent.

### 3.2 Nonlinear Spiking Attention Denoiser (NSAD)

In the attention layers of existing spiking transformers (Zhou et al., 2023), a timestep-wise spiking attention map is generated by multiplying the outputs of the query neuron array ($Q$) with those of the key neuron array ($K$). Each entry in this map corresponds to a pairing of query and key neurons, where a one-to-one spatial correspondence is maintained.

#### 3.2.1 Denosing with Element-wise Nonlinear Transformation

Recognizing the central role of spiking attention maps, we propose a learnable hashmap-based Nonlinear Spiking Attention Denoiser (NSAD) to improve the overall transformer performance. NSAD serves the dual-purpose of denoising a given computed attention map, and equally importantly, introducing efficient element-wise nonlinear transformation to enhance expressive power.

Firstly, it's important to note that a nonzero value in the attention map signifies the simultaneous activation of one or multiple query-key neuron pairs. However, the computed spike-based attention maps are not necessarily devoid of noise, and the existing spiking transformers lack efficient noise suppression mechanisms.

Secondly, it has been shown that applying row or column-based nonlinear softmax operations to attention maps improves performance in ANN-based transformers. However, softmax induces exponential operations and non-local memory access and data summations, which are costly and not hardware-friendly (Dao et al., 2022).

#### 3.2.2 Learning the Nonlinear Spiking Attention Denoiser

The proposed nonlinear spiking attention denoiser (NSAD) offers an efficient solution to addressing the above issues via element-wise hashmap-based nonlinear transformation without non-local memory access and computation, as illustrated in Figure 4.

Each head in a transformer with $H$ heads may have unique focuses and parameter distribution Naseer et al. (2021). As such, we establish a small hashmap $AD^{<h>}$ with $d = D/H$, say $d = 32$, entries dedicated to each head $h \in [1..H]$. Each entry in $AD^{<h>}$ is indexed (addressed) by a specific integer value falling within the range of possible attention values of 0 and $d$, i.e., $AD^{<h>}[s]$ specifies the integer value to which all entries with value $s$ in the attention map associated with head $h$ are transformed to. In other words, the specific $(i,j)$ entry in head-h's attention map, i.e., $S^{<h>}[i,j]$ is transformed to the $AD^{<h>}[S^{<h>}[i,j]]$.

Since NSAD produces nonlinear transformed denoised maps using simple integer-based lookups of small hashmaps, it is computationally efficient and hardware-friendly. For a block of 12-head attention, only $12 \times 32 = 384$ integer values need to be stored in the hashmaps while there are $1.77M$ block-level weight parameters. The complexity of computing a denoised attention map is $O(N \times N)$ per head, which can also be easily parallelized on hardware. This is in sharp contrast to the overall complexity of $O(12ND^2 + N^2D)$ for the block.

We streamline the learning of NSDA as part of gradient-based optimization of the transformer as follows. Instead of directly optimizing each value stored in the hashmaps as an independent integer parameter, we instead impose a proper structure in the desired nonlinear donoising characteristics. As shown in Figure 4, for each head, we define a parameterized continuous-valued one-dimensional nonlinear mapping function

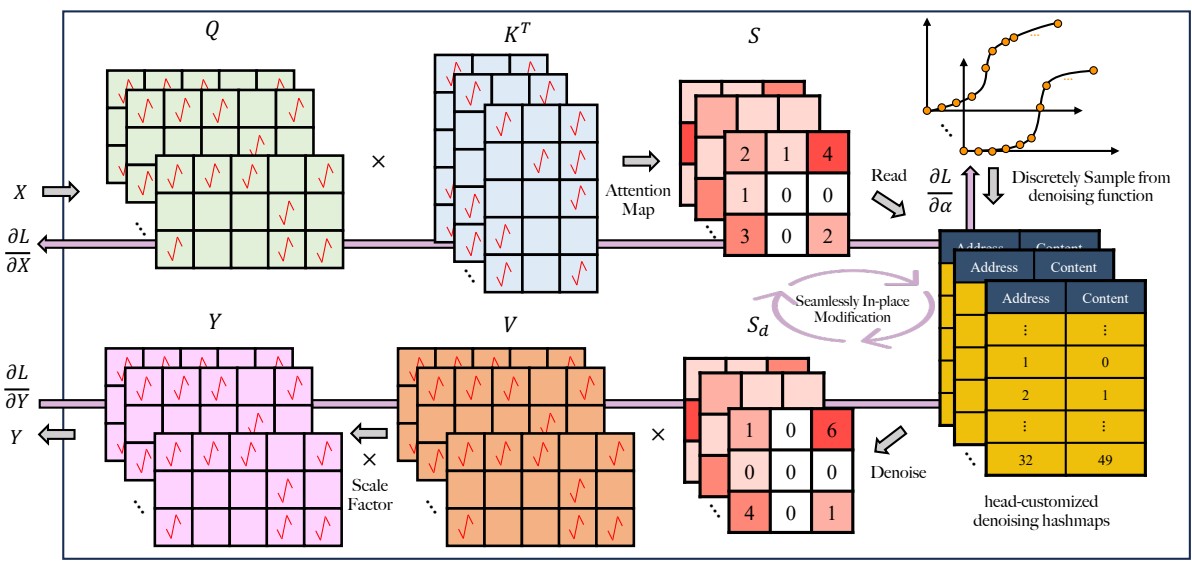

Figure 4: Multi-head hashmap-based Nonlinear Spiking Attention Denoiser (NSAD).

$f_{AD}^{<h>}(\cdot)$ during training, which is parameterized by learnable parameters $\{a_h, b_h, c_h, d_h, e_h, u_h\}$:

$$f_{AD}^{<h>}(s) \quad = \quad \mathbb{1}(s > u_h)g_h(s) \tag{6}$$

$$g_h(s) \quad = \quad \left(a_h s + b_h(s - c_h)^2 + \frac{d_h}{1 + e^{s - e_h}}\right) \tag{7}$$

$f_{AD}^{<h>}$ maps a given input $s$ to 0 if it falls under the denoising threshold $u_h$. Otherwise, $s$ is mapped to $g_h(x)$, which is a weighted sum of a parameterized linear, shifted quadratic, and shifted sigmoid function. This choice of $g_h(\cdot)$ makes it possible to capture three distinct nonlinear characteristics: uniform mapping via the linear function, amplification or suppression of large values of $s$ through the quadratic and sigmoid functions, respectively, or a combination of the three.

Each integer entry $AD^{<h>}[s]$ in the hashmap is obtained by mapping $s$ using $f_{AD}^{<h>}$ and rounding to the nearest integer: $AD^{<h>}[s] = \lfloor f_{AD}^{<h>}(s) \rceil$. While nonlinear denoising of the attention map is applied based on the hashmap $AD^{<h>}$, the continuous parameterization of $f_{AD}^{<h>}(\cdot)$ is optimized by the gradient optimization.

## 4 Experiments

We assess the performance of our `DS2TA` spiking transformer by comparing it with existing SNN networks using various methods trained from scratch and the recent spiking transformer model employing spatial-only attention (Zhou et al., 2023) on several static image datasets such as CIFAR10 and CIFAR100 in Section 4.1, and dynamic neuromorphic datasets such as CIFAR10-DVS and DVS-Gesture in Section 4.2, commonly adopted for evaluating spiking neural networks. We perform our experiments using the Spikingjelly (Fang et al., 2023) frameworks supporting an activation-based gradient surrogate training for SNNs. We further make a sparsity analysis in Section 4.3 and ablation study in Section 4.4.

### 4.1 Results on Static Classification Datasets

**CIFAR10/100** The CIFAR dataset (Krizhevsky, 2009) contains 50,000 training images and 10,000 testing images, assigned into 10 or 100 categories, respectively, and are commonly adopted for testing directly trained SNNs. The pixel resolution of each image is $32 \times 32$. A spiking patch splitting module consisting of 4

| Model | Architecture | Parameters | Timesteps | CIFAR-10 Accuracy | CIFAR-100 Accuracy |
|---|---|---|---|---|---|
| Hybrid training(Rathi et al., 2020) | VGG-11 | 9.27M | 125 | 92.22% | 67.87% |
| STBP(Wu et al., 2018) | CIFARNet | 17.54M | 12 | 89.83% | - |
| TSSL-BP(Zhang & Li, 2020) | CIFARNet | 17.54M | 5 | 91.41% | 74.00% |
| STBP-tdBN(Zheng et al., 2021a) | ResNet-19 | 12.63M | 5 | 92.92% | 74.47% |
| TET(Deng et al., 2022) | ResNet-19 | 12.63M | 4 | 94.44% | 74.47% |
| DT-SNN(Li et al., 2023) | ResNet-19 | 12.63M | 4 | 93.87% | 73.48% |
| Diet-SNN(Rathi & Roy, 2020) | ResNet-20 | 0.27M | 5 | 92.54% | 64.07% |
| Spikformer*(Zhou et al., 2023) | ViT-4-384 | 9.32M | 4 | 94.19% | 76.05% |
| DS2TA | ViT-4-384 | 9.32M | **4** | **94.92%** | **77.47%** |

Table 1: Comparison of DS2TA with other SNNs on CIFAR-10 and CIFAR-100

sequential Conv+BN+MaxPool blocks with gradually improving channels to embedding dimensions, and a relative position generator to patchifies each image into 64 tokens with each token packing 4×4 pixels. In the experiment, we employ a batch size of 256 and adopt the AdamWKingma & Ba (2014) optimizer to train the baseline spikformer model and our DS2TA transformer over 300 epochs, and compare them with several prior SNN works. We employ a standard data augmentation method, such as random augmentation, mixup, or cutmix for a fair comparison. The learning rate is initialized to 0.001 with a cosine learning rate scheduler. We initialize denoising threshold $u_h = 3$ in NSAD for all attention heads; we initialize temporally attenuated attention window size $T_{AW} = 3$ for all neurons and decay exponent $\tau = 4$ for all layers.

We compare the performance of our DS2TA model on CIFAR-10 and CIFAR-100 with a baseline spiking transformer model and other CIFARNet, VGG-11 and ResNet-19 SNNs as shown in Table 1, where the ∗ subscript highlights our reproduced results. The transformer's architecture incorporates 384-dimensional embedding and four encoders (ViT-4-384). DS2TA gains a significant improvement on top-1 accuracy on CIFAR10, improving the baseline spiking transformer's accuracy from 94.19% to 94.92% when executing over 4 time steps. DS2TA obtains a significant accuracy improvement of 0.50% or more over ResNet-19, and improves the accuracy by 2.38% of the ResNet-20 model with 5 timesteps. On CIFAR-100, based on the same ViT-4-384 architecture, DS2TA gains noticeable accuracy improvements over these SNNs and the baseline spiking transformer when executed over 4 time steps. Compared with the baseline spiking transformer model, DS2TA improves by 1.42% on top-1 accuracy; compared with ResNet-19 SNNs, it improves by 3.00% to 3.99% on different training methods, and it improves by 13.4% over ResNet-20 DietSNN.

## 4.2   Results on Dynamic Neuromorphic Datasets

| Model | Architecture | Parameters | CIFAR10-DVS | | DVS-Gesture | |
|---|---|---|---|---|---|---|
| | | | Timesteps | Acc | Timesteps | Acc |
| Rollout(Kugele et al., 2020) | 5-layer CNN | 0.5M | 48 | 66.5% | 240 | 97.2% |
| BNTT(Kim & Panda, 2020) | 6-layer CNN | - | 20 | 63.2% | - | - |
| SALT(Kim & Panda, 2021) | VGG-11 | 9.27M | 20 | 67.1% | - | - |
| PLIF(Fang et al., 2021a) | VGG-11 | 9.27M | 20 | 74.8% | 20 | 97.6% |
| tdBN(Zheng et al., 2021b) | ResNet-19 | 12.63M | 10 | 67.8% | 40 | 96.9% |
| NDA(Li et al., 2022) | ResNet-19 | 12.63M | 10 | 78.0% | - | - |
| DT-SNN(Li et al., 2023) | ResNet-19 | 12.63M | 10 | 74.8% | - | - |
| Spikformer*(Zhou et al., 2023) | ViT-2-256 | 2.58M | 10 | 78.9% | 10 | 93.75% |
| DS2TA | ViT-2-256 | 2.58M | **10** | **79.1%** | **10** | 94.44% |

Table 2: Comparison of the DS2TA spiking transformer with other SNNs on CIFAR10-DVS and DVS-Gesture.

**CIFAR10-DVS and DVS-Gesture** CIFAR10-DVS (Li et al., 2017) is a neuromorphic dataset containing dynamic spike streams captured by a dynamic vision sensor camera viewing moving images from the CIFAR10 datasets. It contains 9,000 training samples and 1,000 test samples. The Dynamic Vision Sensor (DVS)

Gesture dataset Amir et al. (2017) consists of 11 gestures from multiple human subjects as seen through a dynamic vision sensor, an event-based camera that responds to localized changes in brightness. For images from the two neuromorphic datasets, the image size is 128×128, we adapt a 16×16 patch size. We utilize a ViT-2-256 as the backbone, which indicates utilizing two blocks with an embedding dimension of 256. The attention head number for both datasets is set to 16. We employ an AdamW optimizer and a batch size of 256. The learning rate is 0.001 with a cosine decay scheduler. The initial denoising threshold $u_h$ is set to 2 in NSAD; the temporally attenuated attention window $T_{AW}$ is set to 3 for all 16 heads; the decay exponent $\tau$ is set to 2. Neuromorphic data augmentation is applied as in Li et al. (2022); Zhou et al. (2023). The training is performed over 200 epochs.

The classification performances of `DS2TA` and the spiking transformer baseline, as well as other state-of-the-art SNN models are shown in Table 2. `DS2TA` outperforms the existing SNN models under various settings. Compared with the baseline Spikformer transformer model with the same number of 2.58M parameters, it improves by 0.2% and 0.69% the top-1 accuracy on CIFAR10-DVS and DVS-Gesture, respectively. Compared with 12.63M ResNet-19 model, `DS2TA` improves by 1.1%, 4.5%, and 11.3% top-1 accuracy on CIFAR10-DVS using 2.58M parameters, respectively.

## 4.3 Sparsity, Efficiency and Energy Consumption

| Block | Spikformer | | DS2TA | |
|---|---|---|---|---|
| | Sparsity | Energy(nJ) | Sparsity | Energy(nJ) |
| 0 | 74.26% | 7.69 | 96.99% | 0.90 **(88.3%↓)** |
| 1 | 76.82% | 6.93 | 97.82% | 0.65 **(90.6%↓)** |
| 2 | 78.98% | 6.28 | 98.14% | 0.56 **(91.1%↓)** |
| 3 | 82.64% | 5.19 | 98.59% | 0.42 **(91.9%↓)** |

Table 3: Activation sparsity and energy consumption in attention layer under Spikformer and `DS2TA`.

Thanks to our proposed NSAD, the sparsity of the attention map is improved significantly as shown in Figure 3, enabling an opportunity of efficient processing within the attention block. We evaluate the sparsity of spike-based attention maps of our baseline spiking transformer and `DS2TA` model across multiple encoder blocks in ViT-4-384 backbone (CIFAR-100) in Table 3. It's noticeable that the sparsity is improved by 22.73%, 21.00%, 19.16% and 15.95% across four attention maps when it goes from shallow to deep. The activations across many attention heads in `DS2TA` are extremely sparse, and have such opportunities to be pruned completely, introducing a structured sparsity. We build an energy consumption model by $Energy = E_{AC} \times \#OPs \times (1 - sparsity)$ for attention-map related computations. The energy consumption for attention map-related computation is trimmed down by 88.3%, 90.6%, 91.1%, and 91.9%, for blocks ranging from shallow to deeper, respectively.

## 4.4 Ablation Study

| Model | NSAD | ST-attention | Architecture | Timesteps | CIFAR-10 Accuracy | CIFAR-100 Accuracy |
|---|---|---|---|---|---|---|
| Spikformer | × | × | ViT-4-384 | 4 | 94.19% | 76.05% |
| DS2TA | ✓ | × | ViT-4-384 | 4 | 94.64% | 77.19% |
| | ✓ | ✓ | ViT-4-384 | 4 | **94.92%** | **77.47%** |

Table 4: Ablation study on different elements of `DS2TA` on static datasets.

**Proposed key elements in DS2TA.** We analyze the effect of each proposed key element in the `DS2TA` on CIFAR10 and CIFAR100 in Table 4. Overall, the proposed two key techniques can incrementally lead to further performance improvements of the spiking transformer backbones. Under the mode of only enabling NSAD, the top-1 accuracy is improved by 0.45% and 1.14% on CIFAR10 and CIFAR100, respectively; after

adding TASA, the top-1 accuracy is incrementally improved by 0.28% and 0.28% on CIFAR10 and CIFAR100, respectively.

## 5 Conclusion

In this work, we introduce `DS2TA`, a denoising spiking transformer that incorporates parameter-efficient attenuated spatiotemporal attention. We place a particular emphasis on exploring spatiotemporal attention mechanisms within spiking transformers. The mechanism enables spiking transformers to seamlessly fuse spatiotemporal attention information. Furthermore, we introduce a non-linear denoising technique for attenuating the noise and enhancing expressive power of spiking attention maps based on the proposed multi-head hashmap-based denoising mechanism with little extra computational overhead. Moreover, denoising significantly improves sparsity in attention maps, improves computational efficiency, and reduces energy dissipation. Our `DS2TA` spiking transformer outperforms the current state-of-the-art SNN models on several image and neuromorphic datasets. We believe that the presented work provides a promising foundation for future research in the domain of computationally efficient high-performance SNN-based transformer models.

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
