# OpenReview forum: "DS2TA: Denoising Spiking Transformer with Attenuated Spatiotemporal Attention"
_TMLR — Rejected by TMLR_

### Review · Reviewer_N4wD · 2024-07-19

**Summary Of Contributions:**

In contrast to existing spiking transformers that exhibit "spatial-only" attention, such as those by Zhou et al. (2023), DS2TA moves beyond this limitation. It introduces temporally attenuated spatiotemporal attention (TASA) that considers not only spatial but also temporal
correlations in input firings when computing queries, keys, values and final output of each attention block, providing a more comprehensive approach to attention.

DS2TA is designed to facilitate parameter-efficient computation of spatiotemporal attention via a technique called Attenuated Temporal Weight Replica. This approach dramatically reduces the number of temporally-dependent synaptic weights employed in TASA. This efficiency contributes to the overall optimization of the transformer, enhancing its scalability and resource utilization.

DS2TA utilizes efficient hashmap-based nonlinear denoisers with learnable nonlinearity to enhance the robustness and expressive power of spatiotemporal attention maps, thereby further improving performance.

**Audience:**

Yes

**Claims And Evidence:**

Yes

**Requested Changes:**

See the weakness.  In section 3.1.2, it is better to provide more details or discussions for why the proposed replica method is parameter-efficient or reduces parameter number. It is better to have some comparisons for parameter counts or complexity.

In the ablation study, it seems that the denoiser makes most of the contributions for the improvements, while the other parts such as spatiotemporal attention contributes marginally. It is better to discuss more about this, or show the performance with only spatiotemporal attention and no denoiser, to check this problem.

 It is better to extend the model architecture with less/more blocks, or few/more channels to develop a model family with different model size and model performance. It can demonstrate the general performance of this model architecture under different model sizes.

**Strengths And Weaknesses:**

Strength:

It introduces temporally attenuated spatiotemporal attention (TASA) that considers not only spatial but also temporal correlations in input firings when computing queries, keys, values and final output of each attention block, providing a more comprehensive approach to attention.  This new mechanism allows DS2TA to attend to dependencies taking place in both time and space, and provides a means for fully exploring the spatitemporal computing power of spiking neurons under the context of transformer models, as shown in Figure 2.


The proposed NSAD improves the sparsity of the attention map significantly as shown in Figure 3, enabling an opportunity of efficient processing within the attention block.  In Table 3. It’s noticeable that the sparsity is improved by 22.73%, 21.00%, 19.16% and 15.95% across four attention maps when it goes from shallow to deep. The activations across many attention heads in DS2TA are extremely sparse, and have such opportunities to be pruned completely, introducing a structured sparsity.  The energy consumption for attention map-related computation is trimmed down by 88.3%, 90.6%, 91.1%, and 91.9%, for blocks ranging from shallow to deeper, respectively.

Extensive experimental evaluations on various static image and dynamic neuromorphic datasets, consistently demonstrate the superior performance of the DS2TA architecture in comparison to prior spiking transformer approaches.

weakness:

It claims to propose attenuated temporal weight replica to  dramatically reduce the number of temporally-dependent synaptic weights. In section 3.1.2, it is  better to provide more details or discussions  for why  the proposed replica  method is parameter-efficient or reduces parameter number.  It is better to have some comparisons for parameter counts or complexity.

In the ablation study, it seems that the denoiser makes most of the contributions for the improvements, while the other parts such as spatiotemporal attention contributes marginally. It is better to discuss more about this, or show the performance with only spatiotemporal attention and no denoiser, to check this problem.

The model  structure of the proposed method is  quite fixed in experiments. For example, in table 1 or 2, the proposed method only evaluates one single model architecture. It is better to extend the model architecture with  less/more blocks, or few/more channels to develop a model family with different model size and  model performance. It can demonstrate the general performance of this model architecture  under different model sizes.

---

### Review · Reviewer_fFmT · 2024-08-02

**Summary Of Contributions:**

In this paper, the authors propose an architecture called the DS2TA which combines a number of features with Spikformer, specifically including a temporal attention window, attenuation of temporal weights, denoising with a learnable hashmap. They show empirical results on a number of tasks and ablation studies.

**Audience:**

Yes

**Broader Impact Concerns:**

None.

**Claims And Evidence:**

No

**Requested Changes:**

- Critical: Add ablation with binarized neurons instead of spiking neurons and explanation of why LIF neurons are required at all.
- Critical: Fix the clarity aspects of the paper. E.g. what's the input encoding, loss functions, how is denoising done?,
- Critical: Report multiple runs for all experiments.
- Add estimated energy consumption comparison with (some) other SoTA models.
- There are multiple SoTA models missing in DVS-Gesture comparison e.g. see [1].
- Please explain all the symbols in eq. 2 & 3.

**Strengths And Weaknesses:**

### Strengths

- A number of their methods could potentially improve the efficiency of image transformers on appropriate hardware.
- Their method seems to perform well on a number of benchmarks (although only single runs shown).

### Weaknesses

- It is not at all clear what advantage the use of spiking neurons provides in this model, over simple binarized neurons. The temporal dimension of LIF neurons seems to be an unnecessary addition and there's no evidence that it's useful in any way.
- The paper is heard to read with a lot of repetition and is missing clear and detailed descriptions of various parts of the model.
- The authors only show analytical calculations for energy savings. It is not clear if these models would be more efficient on GPUs.
- No detailed analysis done on whether properties of the model actually match that required for neuromorphic hardware.
- None of the empirical results include results over multiple runs. So it's hard to judge the robustness of their method.
- Models are not compared in terms of estimated energy consumption.

---

### Review · Reviewer_cxrB · 2024-08-06

**Summary Of Contributions:**

This paper proposes a Denoising Spiking Transformer (DS2TA) for efficient image classification. The proposed DS2TA consists of two major components: Spiking Temporally Attenuated Spatiotemporal Attention (TASA) and Nonlinear Spiking Attention Denoiser (NSAD). The TASA combines spike information from multiple steps, and the NSAD is a specially designed self-attention method for spiking networks. The authors performed experiments on CIFAR-10, CIFAR-100, CIFAR-10-DVS, and DVS-Gesture datasets. The result shows similar accuracy to other CNN-based SNN works.

**Audience:**

Yes

**Broader Impact Concerns:**

There is no broader impact concern for the paper.

**Claims And Evidence:**

No

**Requested Changes:**

1. Please address the weaknesses above with additional clarification, experiments, and benchmark studies.

2. As stated in Weakness Point 5, the hardware estimation result in Table 3 is questionable. Can the authors explain in more detail how the numbers are computed?

3. Do the authors use any hardware-efficient implementation for TASA to reduce the number of operations? How does the size of the window in TASA influence the final result?

4. Based on the ablation study shown in Table 4, the contribution of TASA is minimal in accuracy improvement. Can the authors explain more about this result and argue why TASA is still needed?

5. Why does the proposed method perform poorly on the DVS-Gesture dataset?

**Strengths And Weaknesses:**

Strength

1. The paper proposed a self-attention method specially designed for SNN with binary events. Additionally, the proposed self-attention method avoids using SoftMax, increasing the hardware efficiency.

2. The proposed method significantly increases the activation sparsity in the attention compared to spikeformer.

Weakness

1. The proposed TASA introduces additional computational and memory costs if implemented in the same way as in Equation 1 of the paper. Firstly, the temporal attention window requires buffering spikes from previous steps and attenuated weight matrixes, increasing the memory overhead. Secondly, every step requires additional computation for the spikes in the temporal attention window, bringing additional computation overhead. The reviewer thinks TASA can be implemented efficiently by changing the computation sequence in Equations 1 and 4. If the authors adopt an efficient implementation of TASA, they need to make it clear in the text and include it in the paper's main text.

2. The main argument for spatial-temporal learning is invalid under the static visual tasks. The tasks selected in the experiments (CIFAR-10, CIFAR-100, CIFAR10-DVS) do not need temporal information for accuracy prediction. Although CIFAR10-DVS is an event-based dataset, it is generated from static images and has little temporal information. The DVS-Gesture dataset is the only one that may need temporal information. However, the proposed method produces low accuracy for DVS-Gesture. If the authors want to claim that spatial-temporal learning truly works, a more complex dataset with rich temporal information is needed in the experiments. For example, [1] uses ViT with LSTM for efficient spatial-temporal learning for event-based object detection datasets.

3. The baseline result for Spikeformer is lower than the original result listed in the source paper. For example, Spikeformer gets 81.4% on CIFAR10-DVS in their paper, but the authors reported 78.9% in this paper. Spikeformer gets 98.9% on DVS-Gesture in their paper, but the authors reported 93.75% in this paper. Additionally, Spikeformer also evaluated the ImageNet dataset, which this paper lacks.

4. The paper lacks comparisons with efficient transformer network algorithms besides the spiking neural network approach. Since the proposed method targets general vision problems, the authors need to compare the proposed method with other efficient transformer designs. For example, many efficient ViT algorithms exist using quantization [2, 3] and sparsity [4, 5]. A thorough comparison of computational cost, memory cost, and accuracy is needed to prove the effectiveness of the proposed method.

5. The energy estimation in the paper is oversimplified and cannot accurately reflect the cost of the proposed algorithm. As stated in Section 4.3, the energy estimation of the network is based solely on the ACC energy cost for each synaptic operation. Firstly, it is unclear if all operations in the network are included in the computation. For example, the energy cost in Table 3 seems proportional to the sparsity. However, based on the first point of weakness, the proposed method has additional overheads compared to regular spikeformers. Secondly, network computation is typically dominated by additional memory access costs, especially for irregular network designs proposed here. Ignoring the memory access costs will create an unfair advantage for the proposed work.

[1] Recurrent Vision Transformers for Object Detection with Event Cameras, CVPR 2023

[2] Post-Training Quantization for Vision Transformer, NeurIPS 2021

[3] Q-ViT: Accurate and Fully Quantized Low-bit Vision Transformer, NeurIPS 2022

[4] Chasing Sparsity in Vision Transformers: An End-to-End Exploration, NeurIPS 2021

[5] SparseViT: Revisiting Activation Sparsity for Efficient High-Resolution Vision Transformer, CVPR 2023

---

### Decision · Action_Editor_fY5w · 2024-09-10

**Recommendation:** Reject

**Comment:**

The claims are not supported, as stated by all reviewers.

**Audience:**

The audience is appropriate but the content of the paper is not.

**Claims And Evidence:**

All reviewers unanimously stated that the paper did not provide support for the claims it made. The authors did not respond to this criticism.